

# Transport of aerosols over the French Riviera - Link between ground-based lidar and spaceborne observations

Patrick Chazette, Julien Totems and Xiaoxia Shang[*]

Laboratoire des Sciences du Climat et de l'Environnement (LSCE), CEA-CNRS-UVSQ, UMR 8212, Gif-sur-Yvette, France
*Now at: Finnish Meteorological Institute, P.O. Box 1627, 70211, Kuopio, Finland

*Correspondence to*: Patrick Chazette (patrick.chazette@lsce.ipsl.fr)

**Abstract.** For the first time, a backscatter $N_2$-Raman lidar has been deployed on the western part of the French Riviera to investigate the vertical aerosol structure in the troposphere. This lidar system, based at the AERONET site of Toulon-La Garde, performed continuous measurements from 24 June to 17 July 2014, within the framework of the multidisciplinary programme
Mediterranean Integrated Studies at the Regional and Local Scales (MISTRALS). By coupling these observations with those of the spaceborne instruments Cloud-Aerosol LIdar with Orthogonal Polarization (CALIOP), Spinning Enhanced Visible and InfraRed Imager (SEVIRI) and Moderate Resolution Imaging Spectroradiometers (MODIS), the spatial extents of the aerosol structures are investigated. The origins of the aerosol plumes are determined using back trajectories computed by the Hybrid Single Particle Lagrangian Integrated Trajectory (HYSPLIT). This synergy allowed to highlight plumes of particulate
pollutants moving in the low and medium free troposphere (up to ~ 5 km above the mean sea level) towards the French Riviera. This pollution originates from the Spanish coast, and more particularly from Costa Blanca (including Murcia) and Costa Brava/Costa Daurada (including Barcelona). It is mainly due to traffic, but also to petrochemical activities in these two regions. Desert aerosol plumes were also sampled by the lidar. The sources of desert aerosols have been identified as the Grand Erg Occidental and Grand Erg Oriental. During desert dust events, we highlight significant differences in the optical characteristics, in terms of backscatter to extinction ratio (BER, inverse of the lidar ratio), between the planetary boundary layer, with 0.024
$sr^{-1}$ (~42 sr), and the free troposphere, with 0.031 $sr^{-1}$ (~32 sr). These differences are greatly reduced in the case of pollution aerosol plumes transport in the free troposphere (i.e. 0.021 and 0.025 $sr^{-1}$). Transported pollution aerosols appear as having similar BER to what is emitted locally. Moreover, using the correlation matrix between lidar aerosol extinction profiles as a function of altitude, we find that during transport events in the low free troposphere, aerosols may be transferred into the planetary boundary layer. We note also that the relative humidity, which is generally higher in the planetary boundary layer
(> 80%), is found to have no significant effect on the BER.

## 1   Introduction

The French Riviera region is the most densely populated area of southern France with, as of 2018, about 4.2 million inhabitants in the Provence – Alpes – Côte d'Azur counties bordering the Mediterranean Sea and the Principality of Monaco. The "Greater



Côte d'Azur region" is also the first tourist destination in France after Paris, with 20 million tourists generating over 130 million overnight stays every year, as well as 1.2 million cruise passengers, increasing the traffic pollutants, especially in summer. Between 1990 and 2005, a marked increasing trend for PM10 ambient concentrations was observed in this area, correlated with an increase in airway diseases (Sicard et al., 2010). The impact of local traffic is predominant in the summer season, but

the industries of the Bouche du Rhône area are also large contributors (El Haddad et al., 2013), and the impact of exogenous sources other than Saharan dust is not negligible (Dimitriou and Kassomenos, 2018). In the coastal town of Toulon, centrally located on the French Riviera, Piazzola et al. (2012) have shown during specific events that air masses could be impacted by pollution transported over the Mediterranean. Yet, these source apportionment studies are only based on surface chemical analyses and backtrajectories, which do not take into consideration the complex meteorological environment of the coastline.

In the framework of the multidisciplinary programme Mediterranean Integrated Studies at the Regional and Local Scales (MISTRALS; http://www.mistrals-home.org), in particular for the Chemistry-Aerosol Mediterranean Experiment (ChArMEx, http://charmex.lsce.ipsl.fr) (Mallet et al., 2015), aerosols in the Mediterranean basin have been studied by several authors, whether via their chemical composition (e.g. Cholakian et al., 2018), their optical properties (e.g. Chazette et al., 2016; Granados-Muñoz et al., 2016), their radiative budget (e.g. Nabat et al., 2015; Di Biagio et al., 2016; Sicard et al., 2016), or the

identification of their sources (e.g. Ancellet et al., 2016; Chrit et al., 2018). Among these studies, few were conducted in the atmospheric column above the French Mediterranean coast, which may be subject to aerosol loads of very different origins and chemical compositions. These aerosols directly influence the air quality (e.g. Knipping and Dabdub, 2003), as well as the climate balance of the Western Mediterranean Sea (e.g. IPCC, 2014; Nabat et al., 2015), in different ways depending on their nature and the surface albedo. For instance, during the Hydrological Cycle in the Mediterranean Experiment (HyMeX, also

part of MISTRAL program), the radiative effect of dust aerosols has been shown to have little impact in the rainfall amounts and location over the Western Mediterranean basin (Flamant et al., 2015).

All of these studies were preceded by early campaigns such as that of the European project Mediteranean Dust Experiment (e.g. Hamonou et al., 1999), or even networked observations such as those of the lidar Earlinet network (e.g. Balis et al., 2000; Pappalardo et al., 2004; Papayannis et al., 2008; and more recently Granados-Muñoz et al., 2016). Coupling in situ

measurements and modelling, the vertical structure of the planetary boundary layer under sea-breeze conditions was also invertigated during the ExperimentS to COnstrain Models of atmospheric Pollution and Transport of Emissions (ESCOMPTE, Cros et al., 2004), over the Marseille-Berre area, ~40 km west of the French Riviera.

Little information exists about the description of transboundary transport of aerosols within the free troposphere over the French Riviera. For this reason, a ground-based $N_2$-Raman lidar was installed near this site between 24 June and 16 July, 2014.

The lidar has combined more than 500 hours of continuous operation and has made it possible to carry out a significant study of the aerosol types and origins, in synergy with spaceborne observations and back trajectories modelling. Works performed using lidar measurements over the Balearic Island of Menorca (Chazette et al., 2015) already highlighted long range transports of aerosols coming from forest fires in North America (see also Ancellet et al., 2016), from deserts and, to a lesser extent, from pollution sources located on the Costa Brava (Barcelona). During the summer period, we were not able to establish a clear link



between the polluted air masses passing over Menorca and those reaching the French Riviera, suggesting another pathway and/or other aerosol sources. Moreover, previous studies carried out on the French Riviera were only based on surface observations (e.g. Piazzola et al., 2012) and have not offered the possibility of clearly identifying the origin of aerosols, which will be shown hereafter.

The experimental strategy is developed in Section 2, where the lidar, the signal processing and the main uncertainty sources is presented. The temporal evolutions of aerosol optical properties and the vertical atmospheric structure over the French Riviera, obeserved from ground-based lidar system, are discussed in Section 3. Section 4 is devoted to the meterological conditions during the field campaign. The long range transport of aerosol plumes highlighted from the lidar measurements is described in Section 5, using the coupling with both spaceborne measurements and back trajectory studies. Section 6

summarizes and concludes.

## 2    Strategy

The perimeter of the Western Mediterranean is composed of mountains with elevations generally greater than 1000 m above mean sea level (AMSL). This specific morphology facilitates the recirculation of air masses, and therefore aerosols, via the sea breeze / land breeze cycle, the alternation between the katabatic and anabatic winds, and the guiding of the air masses

circulation in the valleys and along the sea shore. Sea breeze is an effective means of exchange between the marine or continental boundary layer and the free troposphere. The aerosols trapped in the low and medium free troposphere will then be transported over long distances and may arrive over the Mediterranean coasts where they will reach the surface via free or forced convection processes generated by the mountains. This transport often happens in thin sheet-like plumes whose vertical limits are generally marked by discontinuities of the potential temperature gradient (e.g. Dalaudier et al., 1994; Chazette et al.,

2001). The observation of such layers requires a profiler with a high vertical resolution and justified the deployment of a $N_2$-Raman lidar on the French Riviera, close to the AERONET (AErosol RObotic NETwork) site of Toulon-La Garde (43.13556° N, 6.00944° E, 50 m elevation), which is representative of peri-urban conditions. The location of the site is given in Figure 1a. The lidar performed automatic measurements from 24 June to 17 July, 2014. It was remote-controlled from the Paris area. Lidar measurements are investigated together with active and passive remote sensing spaceborne observations, as well as by

back trajectory studies.




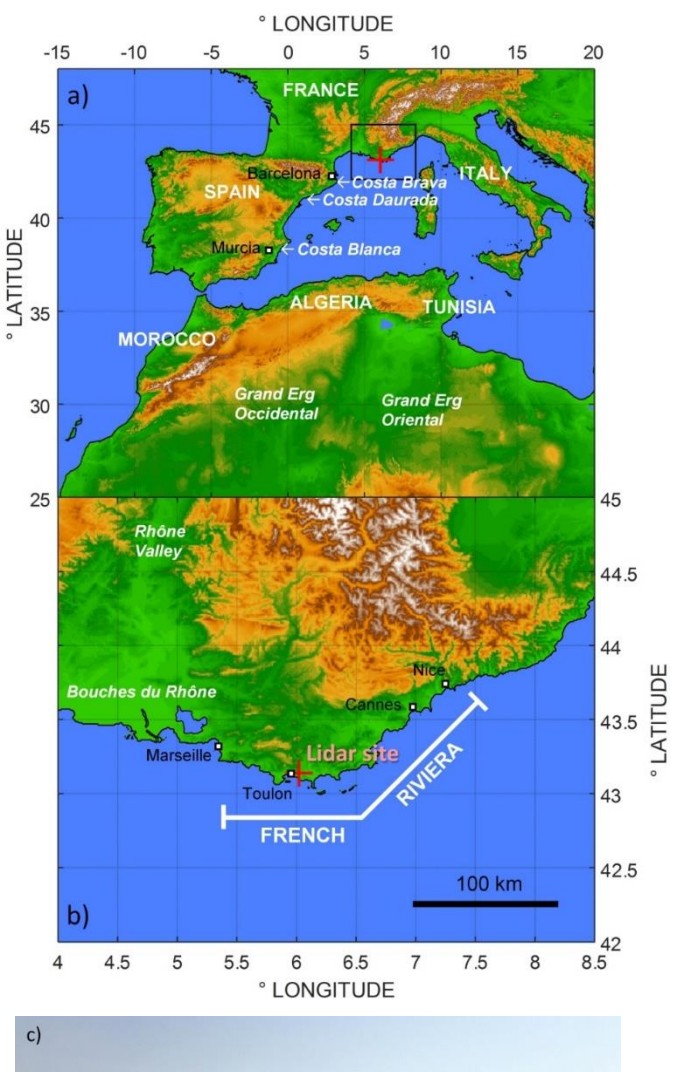

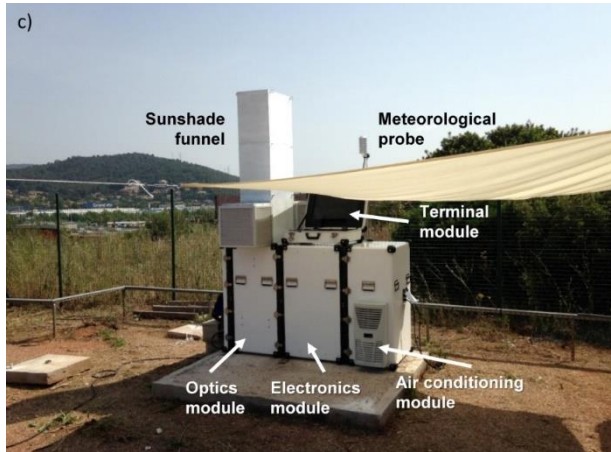

**Figure 1: a) Map of the Western Mediterranean Basin and b) crop showing the location of the lidar at the Toulon-La Garde AERONET site on the French Riviera (elevation data from the GTopo30 DEM, courtesy of USGS). c) Lidar in its confinement during the field experiment.**



## 2.1 The automatic $N_2$-Raman lidar

The $N_2$-Raman lidar LAASURS (Lidar Automatic for Atmospheric Surveys using Raman Scattering) is a research instrument composed of three channels for the parallel and perpendicular polarizations with respect to the laser emission, and the inelastic nitrogen vibrational Raman line of the laser induced atmospheric backscattered signal (Chazette et al., 2014; Chazette et al., 2016). The lidar in its confinement is shown in Figure 1b. The emission energy was 16 mJ at the wavelength of 355 nm and fulfilled eye-safety requirements. The overlap function of the lidar is equal to 1 at distances between 150 and 250 m from the emission. The emission comes from an Ultra® Nd:YAG laser manufactured by Quantel, delivering 6 ns width pulses at the repetition rate of 20 Hz. The detection is carried out by photomultiplier tubes and narrowband filters with a bandwidth of 0.2 nm. The signal acquisition is performed using the PXI technology (http://www.ni.com) with a native vertical sampling of 0.75 m. Note that the acquisition uses a pre-trigger for the correction of the sky background. This avoids effects related to the potential variability of the baseline of analog to digital converters.

## 2.2 Inversion scheme

In order to retrieve the optical properties of aerosols transported in the free troposphere, we used a $N_2$-Raman lidar system coupled with the sunphotometer (e.g. Royer et al., 2011; Chazette et al., 2016) of the Toulon-La Garde AERONET (http://aeronet.gsfc.nasa.gov/) site. Indeed, the signal to noise ratio (SNR) of the daytime lidar profiles is insufficient to use the $N_2$-Raman channel as it presents a low emitted energy. Contrariwise, the $N_2$-Raman channel was used during night time for the retrieval of the cumulative aerosol optical thickness (AOT). Combined with the elastic channel, they lead to the retrieval of the aerosol backscatter and extinction coefficients (ABC and AEC), and to their ratio (Chazette et al., 2016). The backscatter to extinction ratio (BER) is equal to the product of the single scattering albedo and of the probability of a photon being backscattered after an interaction between the laser flux and the atmospheric scatterers. It is the inverse of the lidar ratio (LR) often used in the literature. We prefer to consider hereafter the BER, which has a more direct physical meaning. The linear particle depolarization ratio (PDR) is also retrieved as in Chazette et al. (2012). Calculations are performed with a temporal resolution of at least 30 minutes and a vertical resolution of 30 m to improve the SNR in the middle troposphere (between 6 and 7 km AMSL).

Compared to the inversion scheme proposed in Chazette et al. (2016), where an equivalent BER for the entire aerosol column is justified, the frequent presence of two aerosol layers, one in the marine boundary layer and the other in the free troposphere, led us to develop a multi-layer inversion to evaluate independently the BERs of each aerosol layer and verify the relevance of the first approach leading to an equivalent BER. This method is like the previous one, it uses the partial AOT calculated from the $N_2$-Raman channel for each aerosol layers. The transition altitude between the layers is determined manually and the continuity is ensured by a sigmoid function on a thickness of about 1 km between the two layers. Such an approach is possible, especially for the upper layer, if the SNR is larger than 10. This leads us to perform night time profiles with a time average of 5 hours, between 23:00 and 04:00.



## 2.3 Uncertainties

The main uncertainties sources are discussed in Royer et al. (2011). The relative uncertainty on the $N_2$-Raman-derived cumulative AOT is less than 2% for SNR > 10. The uncertainty in the determination of the equivalent BER is in the range of 4-6 $10^{-3}$ $sr^{-1}$ (10-15 sr in terms of LR). Such a value is very dependent on the SNR, which limits the relevant range of the lidar profile, as shown in Table 2 of Dieudonné et al. (2017). The relative uncertainties on the PDR are close to 10% for the AOTs encountered at 355 nm (AOT> 0.2).

## 3 Lidar observations

### 3.1 Temporal evolution

The temporal evolution of vertical profiles of optical parameters derived from lidar observations are shown in Figures 2 and 3, separated as 2 periods: before and after 7 July. Several days show anomalies of the aerosol load in the free troposphere, whereas others are more common, with a marked signature of the boundary layer cycle. The aerosol layers above 1.5 km AMSL are generally linked to long range transport, especially on 24, 28 June, 1 to 4 and 7 July (high values of AEC in Figure 2b). Over the coastal site, aerosols within the planetary boundary layer (PBL) are mainly from local sources, either sea-spray aerosols generated by breaking waves (e.g. Yoon et al., 2007), or continental component arising from both natural and anthropogenic sources. The relative influence of these aerosol types is modulated by the land/sea breeze cycle (e.g. Piazzola et al., 2012).

The values of PDR are very variable, ranging between ~1% and more than 20%. These highest values are observed in the free troposphere on 24 June and 3-4 July and may be associated with plumes of terrigenous aerosol with non-spherical shapes. In the PBL, we note the existence of vertical streaks which are the signature of thermal up draughts developing during the day. They can lift local terrigenous particles and even pollens to the PBL top. The PDR is greater, but not very high inside these structures (2-3%). Indeed, these aerosols are most probably mixed with a significant quantity of spherical hygroscopic particles from the sea or from local pollution.

Relatively high AOTs are measured over much of the observation period, with values exceeding 0.2 at 355 nm and peak values greater than 0.5. AOT values below 0.2 correspond to undisturbed periods, i.e. without the presence of aerosol layers in the free troposphere. Hereafter, days are tagged as disturbed with an aerosol charge anomaly in the free troposphere, or as undisturbed in the contrary case. It should be noted that, during the second period of the measurement campaign, there is no significant aerosol load in the free troposphere. During the first period (24 June to 7 July), it is rather the opposite, except from 30 June to 1 July. These two days are grouped together with those of the second period in our analysis. Moreover, these two days show a very strong variation of the BER as it is the case in the beginning of the second period, from 8 to 10 July (0.020 $sr^{-1}$ during night time and 0.035 $sr^{-1}$ during daytime). These days are associated with low relative humidity at the ground level: below 50%, as shown in Figure 4. The liquefaction point of the soluble compounds trapped on the aerosol was therefore



probably not reached (e.g. Randriamiarisoa et al., 2006). The strong variations of BER are certainly attributable to the local breeze regime with a stronger marine contribution during the day and therefore larger aerosols. The daytime value is similar to that found by Flamant et al. (1998), which was ~0.040 sr$^{-1}$.



**Figure 2: Temporal evolution between 24 June and 7 July of a) the backscatter to extinction ratio (BER, mean value as a black line and error bar in orange), as well as the aerosol optical thicknesses (AOT) at 355 nm derived from the sunphotometer (red circles) and lidar measurements (blue triangle), b) the vertical profile of the aerosol extinction coefficient (AEC) at 355 nm, and c) the vertical profile of the linear particulate depolarization ratio (PDR) at 355 nm (*PDR*). White time stripes correspond to periods of low and middle clouds. The origins of the main aerosol plumes trapped in the free troposphere are indicated.**

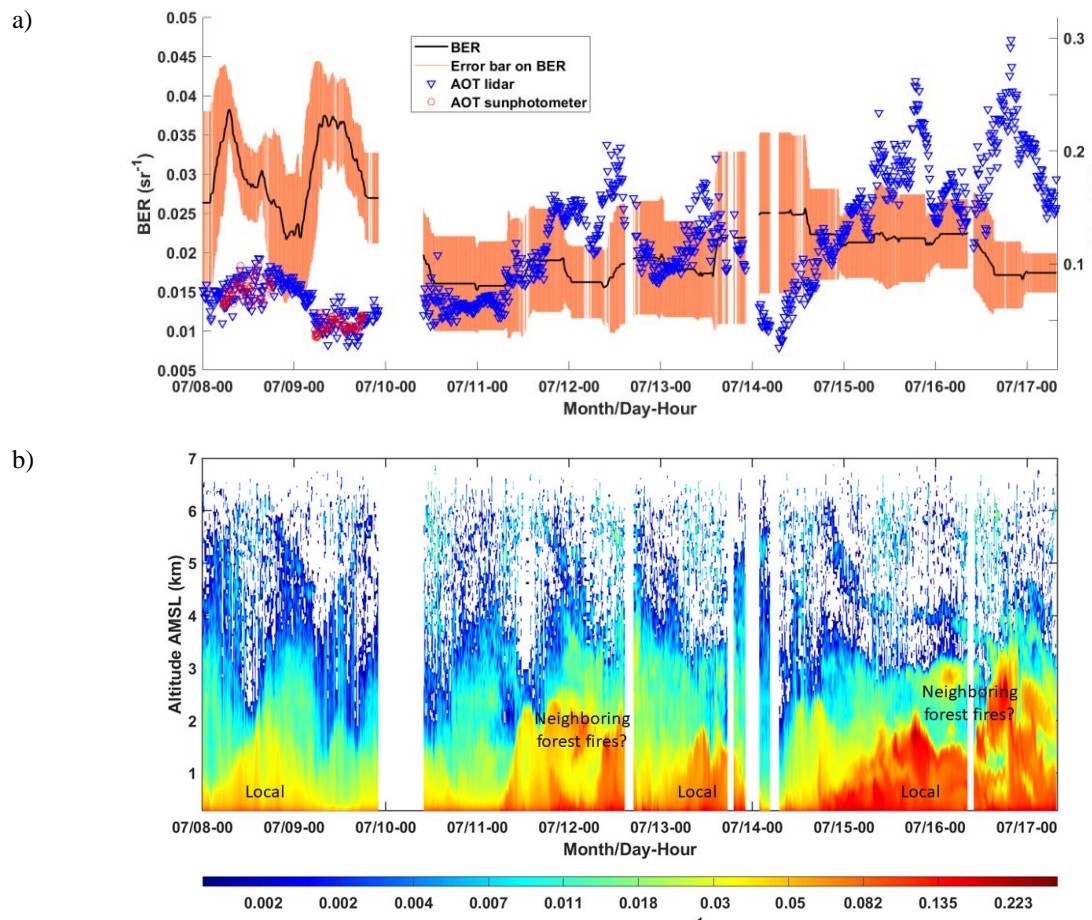





c)

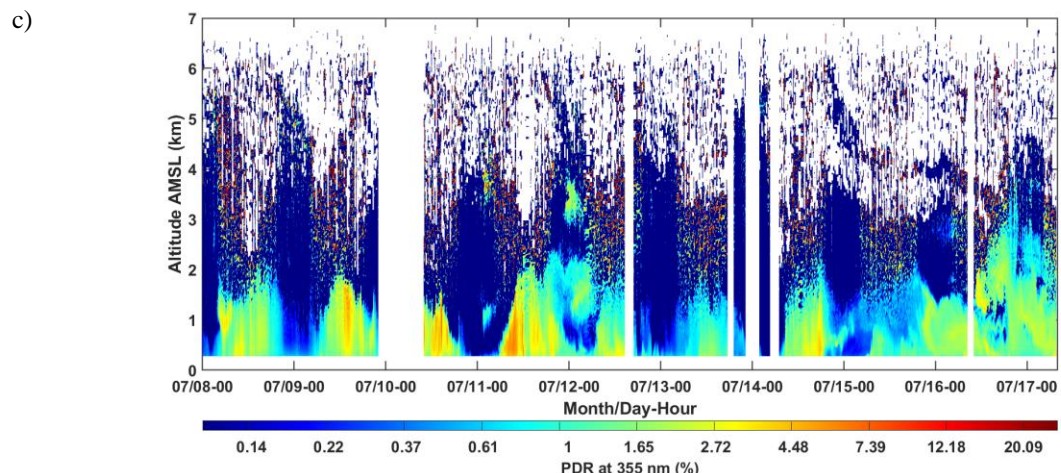

**Figure 3: Same as Figure 2, for the period between 8 and 17 July.**

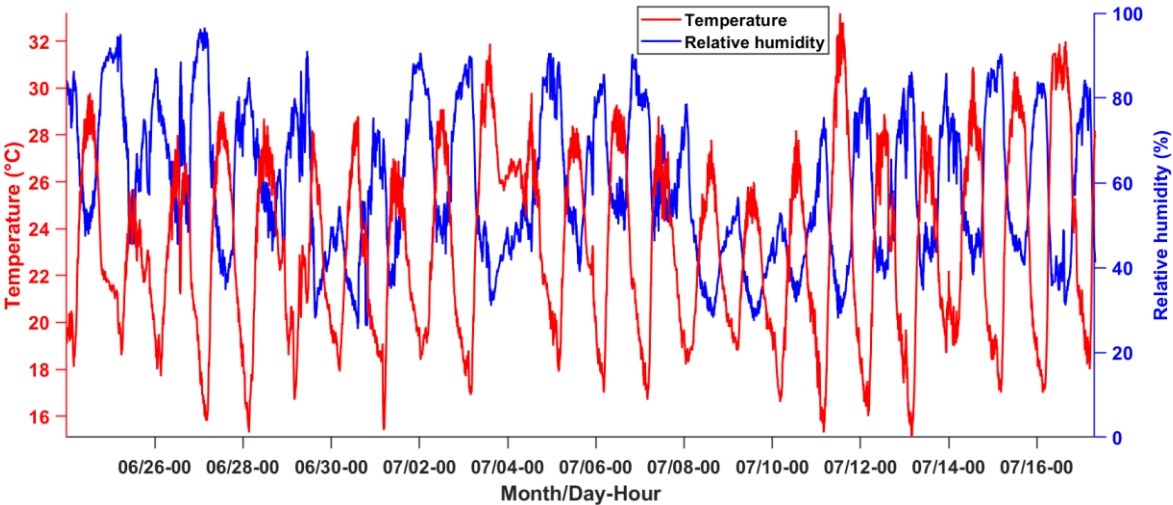

**Figure 4: Temporal evolutions, from 4 June to 17 July 2014, of thermodynamic temperature and relative humidity at 2 m above the ground level.**

5    **3.2    Variability along the altitude**

Before proceeding further, we will check whether or not the hypothesis of a constant BER in the whole atmospheric column is justified. An easy way to verify this is to compare the results on the profiles of the cumulative AOT derived from the inversion of the elastic channel and the one from the $N_2$-Raman channel as in Chazette et al. (2017): the selected BER value is close to the real one if the two profiles coincide.

10    Here, the two-layer method described in Section 2.2 is here applied for night time lidar profile inversion. The profiles of cumulative AOT for the main cases of the campaign are given in Figure 5. We noticed that BER values in the lowermost layers



(0.024 sr$^{-1}$) are only significantly different to the ones found in the upper aerosol layers (0.031 sr$^{-1}$) in the case of 24 June. On the same day, the retrieved column-equivalent BER ranges between 0.017 and 0.033 sr$^{-1}$, with a mean value at 0.027 sr$^{-1}$. Taking uncertainties into account, these values overlap. We can, however, expect an error on total AOT of about 15 to 20%. For the other nights, the hypothesis of one column-equivalent BER value is reasonable, although the nature of aerosols can be different between layers. In our case, there is a strong anthropic component associated with the marine aerosols over the measurement site (Piazzola et al., 2012). These aerosols being hydrophilic, both their size and their complex refraction index can change significantly between the PBL and the free troposphere (Randriamiarisoa et al., 2006). Nevertheless, the work of Raut and Chazette (2008b) has shown that a variation in relative humidity does not modify significantly the value of BER for traffic aerosols. Here, the low vertical variability of BER highlighted supports these findings, given that the relative humidity in the PBL was mainly above 80%, whereas it was below 50% in the free troposphere.

In addition, the study of correlation matrices between altitudes for i) the whole duration of the campaign, and ii) for the undisturbed period, shows that there are correlated aerosol plumes between the PBL and the free troposphere, which can be due to transfers of aerosols between these two layers. Figure 6 gives a graphical representation of the magnitude of the coefficients in the matrices of both the entire measurement period between 24 June and 17 July, 2014, and the non-perturbated period. For undisturbed cases (Figure 6a), the correlation distance (correlation coefficient > 0.6) does not exceed ~2 km, whereas it largely exceeds 3 km for the whole duration of the campaign. This argues for the conclusion that during transport events in the low free troposphere, aerosols may be transferred into the PBL. Simultaneous transport between the marine boundary layer and the free troposphere is unlikely over the large distance separating the emission and the arrival of the aerosol plumes over the French Riviera. The more relevant hypothesis is that the recirculation of air masses along the sea shore due to relief and a modification of average winds in the PBL between undisturbed and disturbed cases explain these transfers. Furthermore, large correlation distances are found in the middle free troposphere (Figure 6b), where aerosol plumes have been transported over a long distance. For undisturbed situations above 5.5 km AMSL, the patchiness of the data is due to the relatively high noise when aerosol scatters are almost absent.





**Figure 5: Vertical profiles of the cumulative aerosol optical thickness (AOT) for six different nights. The lidar profiles are time-averaged between 23:00 the previous day and 04:00 the next day. The inflection levels between the two main aerosol layers are highlighted by a horizontal blue line for each night. The backscatter to extinction ratios (BERs) are given in the legends between brackets for the lower (BER$_{dl}$) and upper (BER$_{ul}$) aerosol layers.**



a)                                                      b)

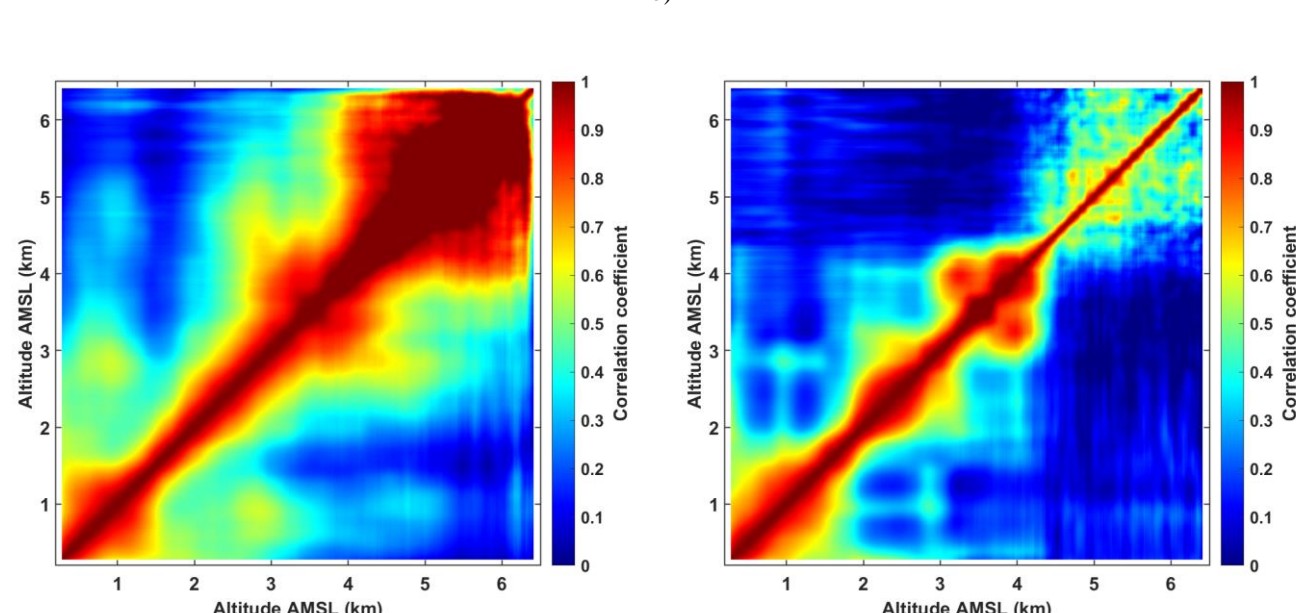

**Figure 6: Lidar-derived correlation matrices for a) the entire measurement period between 24 June and 17 July, 2014, and b) the undisturbed period (without aerosol load anomaly in the free troposphere). These matrices are symmetric by construction.**



## 4    Meteorological conditions

Wind conditions control the transport of aerosols above the French Riviera. To analyse the meteorological conditions during the field campaign over our observation site of Toulon, we use the Era5 reanalyses (https://www.ecmwf.int/en/forecasts/datasets/archive-datasets/reanalysis-datasets/era5) with a horizontal resolution of 0.25 or 0.30°. This dataset is provided by the European Centre for Medium-range Weather Forecasts/integrated forecast system (ECMWF), developed through the Copernicus Climate Change Service (https://climate.copernicus.eu/).

Figure 7 shows the wind direction distribution, computed from Era5 data for the measurement site, at the 975 hPa level (~0.4 km AMSL) during disturbed cases (Figure 7a) and the undisturbed cases (Figure 7b). For the undisturbed cases, winds come from North-West in great majority, and there is little marine contribution. We therefore do not observe a marked see/land breeze cycle at the model scale. In this configuration, one can expect the influence of pollution sources to be linked to road traffic, which is intense in the summer season just north of the measurement site, or of possible biomass fires, very frequent in the backcountry at this period. For disturbed cases, the origin of winds near the surface is much more diverse and reflects the breeze cycle with a significant marine contribution from the south sector during the day, and dominant winds from the east or west along the coast. There is always a west/north-west component from the backcountry. In the lower free troposphere, there are also strong differences between disturbed and undisturbed situations. Figure 8 presents wind distributions at the 700 hPa level (~3 km AMSL). For undisturbed cases (Figure 8b), winds are very predominantly from North/North-West. It is much less distinct during disturbed cases, and presumably multiple contributions are observed from the West / North-West, West / South-West and South / South-West. The different potential contributions are explained hereafter.

On a larger scale, the circulation of air masses advected over the western Mediterranean mostly depends on the relative positions of the Azores and the Siberian highs. It is strongly modulated by lows travelling east over mid latitudes. Depending on the position of these lows, air masses from the Atlantic (over Gibraltar) and from the Sahara can supply the western Mediterranean coast. Figure 9 gives an illustration of the two different configurations predominantly encountered during the measurement campaign.

In Figure 9a, the meteorological situation of 24 June is presented for level 700 hPa of Era5 reanalyses. It corresponds to elevated values of PDR in the lower free troposphere (Figure 2c). Tropical air masses are channelled by the presence of two highs, one over North Atlantic (Azores high) and the other over the Sahara, as well as a weak low over the Iberian Peninsula and a strong low over Scandinavia. This configuration favours the transport of desert dust aerosols over the Mediterranean Sea (Hamonou et al., 1999). It is repeated on 3-4 July, when strong PDR values are also observed. Concerning the aerosol plumes observed on 1, 2, 4 July (Figure 2b), we found a displacement towards the British Isles and a strengthening of the low near Iceland (Figure 9a), and a weakening of the low near Scandinavia. Tropical air masses are then deviated towards Sardinia, and air masses from the eastern Spanish coast are more often advected above the marine boundary layer, which explains the significant decrease of PDR in the observed aerosol layers.





The second meteorological configuration is illustrated in Figure 9b for 28 June. It shows an Atlantic circulation mostly driven by the location of the low over the British Isles and Scandinavia. This circulation favours the arrival of air masses from the Spanish coast over the French Riviera area. The situation is very similar on 7 July, with a deepening of the low on the British Isles and a high on the Baltic countries.

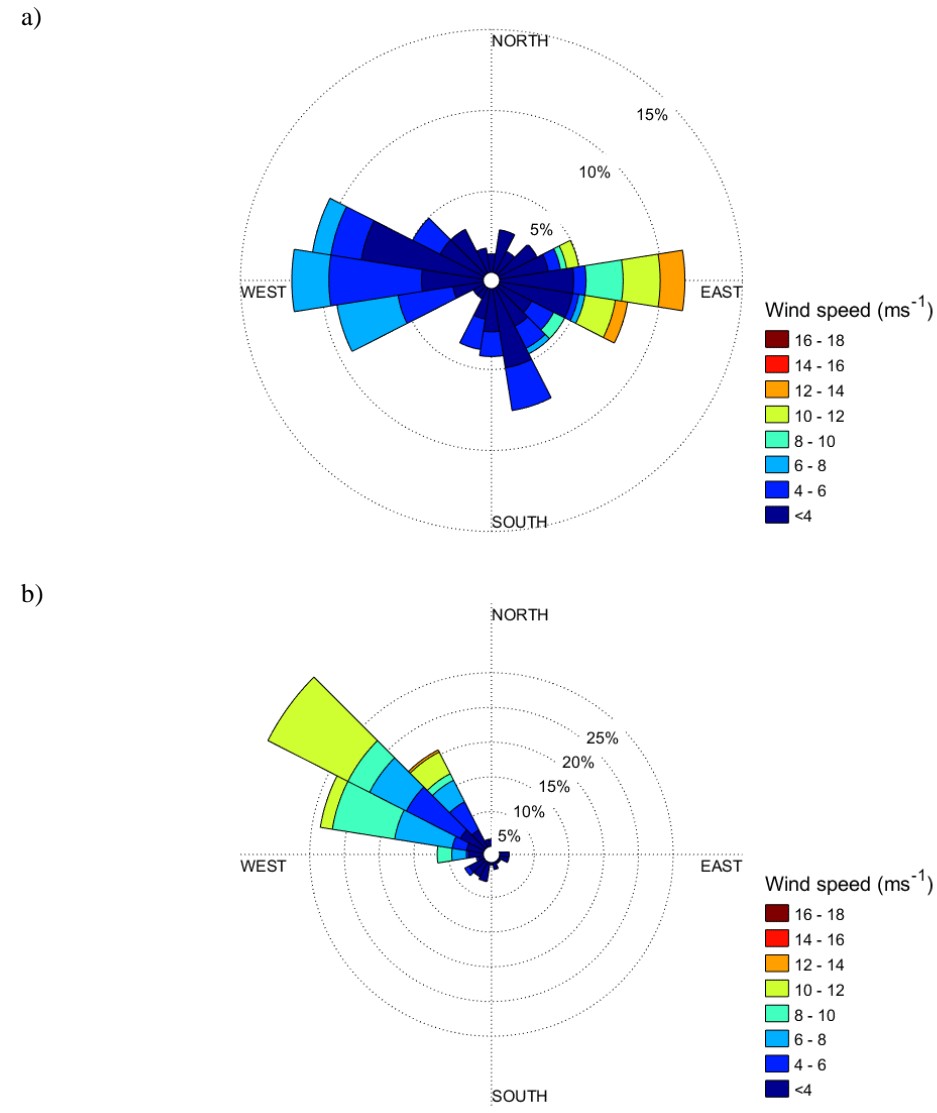

5      **Figure 7: Frequency of counts (%) by wind speed direction at 975 hPa (~0.4 km AMSL) for  a) disturbed cases, days with aerosol anomaly within the free troposphere and b) undisturbed cases, days without anomaly. The hourly data are from the Era5 reanalyzes at 0.25° of horizontal resolution.**



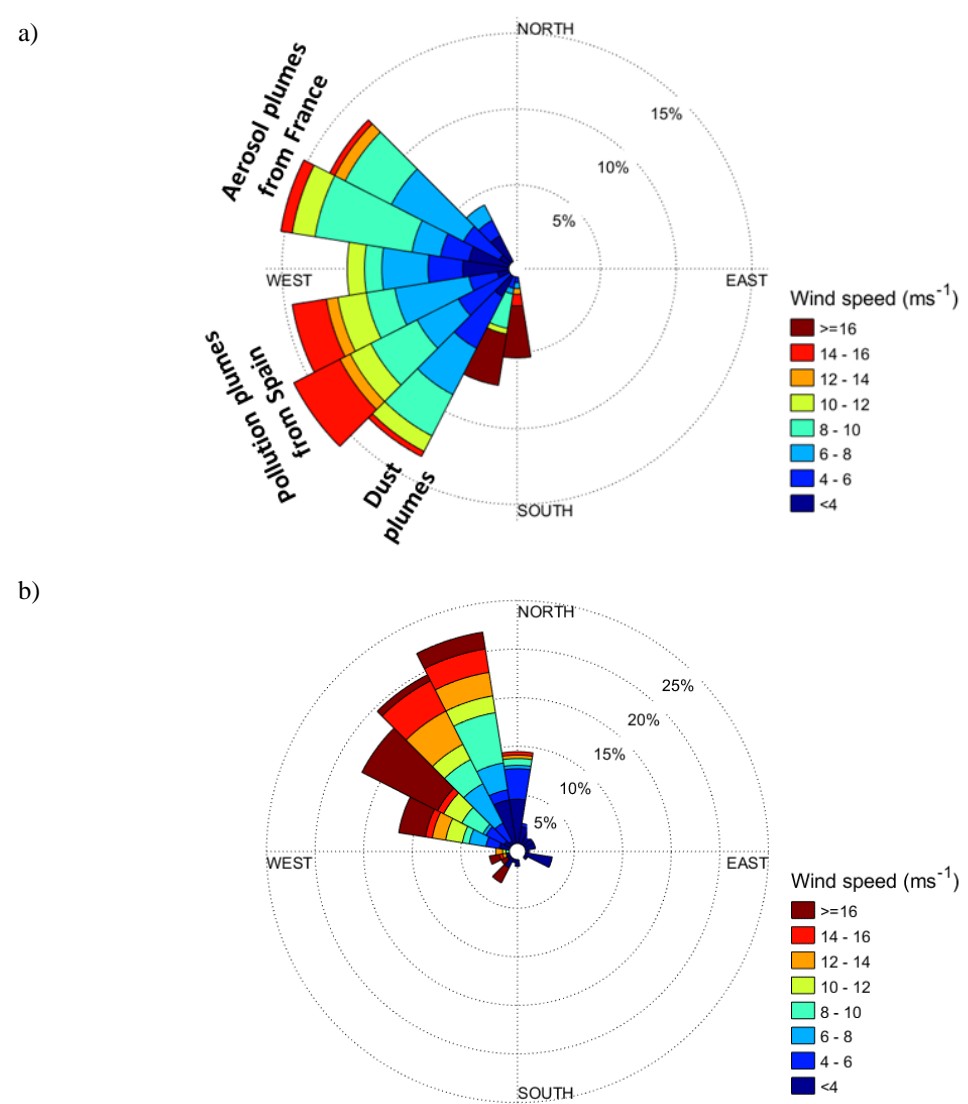

**Figure 8: Same as Figure 7, for the pressure level of 700 hPa (~3 km AMSL).**




a)

b)

**Figure 9: Geopotential altitude from the Era5 reanalyses given for the pressure levels of 700 hPa at 1200 UTC on a) 24 and b) 28 June 2014. The wind field is also shown on each figure. The horizontal resolution is 0.30°.**



## 5    Aerosol transport in the free troposphere

Two types of aerosol transports are described using a joint approach between lidar measurements, spatial observations and back trajectory modelling. To complete and generalize the ground-based lidar measurements, the version 4.10 of level-2 products of the Cloud-Aerosol LIdar with Orthogonal Polarization (CALIOP, Winker et al., (2007)) is used (https://www-calipso.larc.nasa.gov/products/). The horizontal extent of the events over the Western Mediterranean basin is studied using the coupling between the Spinning Enhanced Visible and InfraRed Imager (SEVIRI, https://wdc.dlr.de/sensors/seviri/, Bennouna et al. (2009)) aboard Meteosat Second Generation and the Moderate Resolution Imaging Spectroradiometers (MODIS, http://modis-atmos.gsfc.nasa.gov, Salmonson et al. (1989)) on-board the Aqua and Terra platforms. The back trajectories are computed using the Hybrid Single Particle Lagrangian Integrated Trajectory (HYSPLIT) model (e.g. Stein et al., 2015). The model is initialized using the wind fields of the Global Data Assimilation System (http://www.ncep.noaa.gov/) at 0.5° horizontal resolution and works in its ensemble mode, i.e. 27 back trajectories are computed for each end location. The end-points of the back trajectories are defined using the lidar profiles in Figures 2 or 3. The origin of the aerosol plumes trapped in the free troposphere is highlighted in Figure 8a and discussed hereafter.

### 5.1    Spain's contribution

It is between 28 June and 7 July that aerosol plumes with low depolarization have been observed in the free troposphere (cf. Figure 2), mostly on 28-29 June and 6-7 July. The study of back trajectories shows that most of these plumes originate from the Spanish Mediterranean coast (Figure 10). They take about one day to reach the western part of the French Riviera. We have identified two main contributing regions on the Spanish coast: the region of Barcelona (Costa Brava and Costa Dorada) and the region of Murcia (Costa Blanca). The origins of the plumes are indicated at 700 hPa on the wind rose of Figure 8a and in Figure 2b, and also take into account pollution plumes of weaker magnitude. The contributions of the different sources observed above the French Riviera are not systematically mixed when arriving above the lidar site but are frequently separated over different altitudes. For this reason, back trajectories starting at different altitudes have been performed for the same day. On 28 June, the aerosol plume in the lower free troposphere, around 2 km AMSL, is predominantly from the Murcia region (Figure 10a). AOT values above 0.4 have been recorded at 355 nm wavelength by the local AERONET station of Murcia 2 days prior. The aerosol plumes originating from the region of Barcelona are mostly observed at higher altitude, around 3.5 km (Figure 10b), with similar local AOTs. The visible Ångström exponent is characteristic of pollution aerosols with values over 1.5. The aerosol plume on 7 July originates from the same regions, but with a contribution from Barcelona in the lower free troposphere (Figure 10c), with local AOTs at 355 nm around 0.3 two days prior, and Ångström exponent values close to those of the previous case. The contribution of the Murcia region starts a day earlier and at a higher altitude (Figure 10d), with similar AOTs and Ångström exponents.

The AOT fields at 550 nm derived from MODIS observations are presented in Figure 11 for the two main dates with aerosol plumes in the free troposphere. The main contributing areas are located around Barcelona and specifically around Murcia,



where AOTs above 0.4 are observed. Note that no major forest fire could be identified in the Iberian Peninsula from the MODIS fire product (http://modis-fire.umd.edu/index.php) during the whole measurement period. In Barcelona, road traffic is the foremost cause of pollution, with an urban conglomeration of more than 1.6 million inhabitants (Dall'Osto et al., 2012). The emissions linked to the automobile industry, petro-chemistry, and shipping activity also contribute in this area. As for

Murcia, it is a conglomeration with more than 700,000 inhabitants, in which traffic also has a non-negligible impact, but cannot single-handedly explain the strong signature seen on the MODIS AOT map. It is also a region with intense agriculture, which has been dubbed "Europe's orchard", nonetheless the June period is not suitable for slash-and-burn or muck-spreading. However, the locality of Escombreras, near Carthagena, in the south-east of Murcia, includes a gigantic seaside oil processing complex, with a refinery and a harbor. This area can therefore be a strong emitter of aerosol precursors: excluding desert dust

episodes, MODIS observations often show AOT values above 0.6 in the south of Murcia during summertime. A mix of traffic and industrial emissions can certainly explain the plume observed over Murcia, part of which is transported towards the French Riviera.

There are no CALIOP observations of the Barcelona area. Conversely, there are daytime and night time orbits overflying the region of Murcia, yet none of these orbits are exploitable during the campaign, as they are mostly associated to desert dust

episodes, very frequent in the region. Local aerosols therefore cannot be isolated with good accuracy. In order to obtain the CALIOP classification of the aerosols emitted in the Murcia region (Burton et al., 2013), we have thus extended our search for a coincident night time orbit until August. The choice of a night time orbit is motivated by the need for higher signal to noise ratio. The only day in summer 2014 when CALIOP overflew Murcia and local aerosols were recorded is 25 August. Figure 12 presents the MODIS image and the corresponding CALIOP orbit. The two observations are 12 hours apart. The

MODIS-derived AOT values are close to those on Figure 11. The CALIOP aerosol classification scheme indicates mostly polluted dust with a BER of 0.018 sr$^{-1}$ ($LR$ = 55 sr), matching well the values identified in the aerosol plumes from the ground-based lidar (Figure 5).

The pollution plumes from Barcelona and Murcia can be injected in the free troposphere when the warm continental air mass is advected over the colder water of the Mediterranean Sea. This process can lead to the injection of pollution aerosols up to

altitudes exceeding 4 km AMSL, as observed on Figure 2b. Such plumes can then be seamlessly transported towards and above the French Riviera. These pollution particles are finally eliminated mostly by rainfall and will reach the surface and water streams. The probable presence of black carbon, as identified by Chrit et al. (2018), will favour the trapping of solar energy in the aerosol layer and induce local heating, which is known to modify the balance of the low and middle troposphere (e.g. Raut and Chazette, 2008).





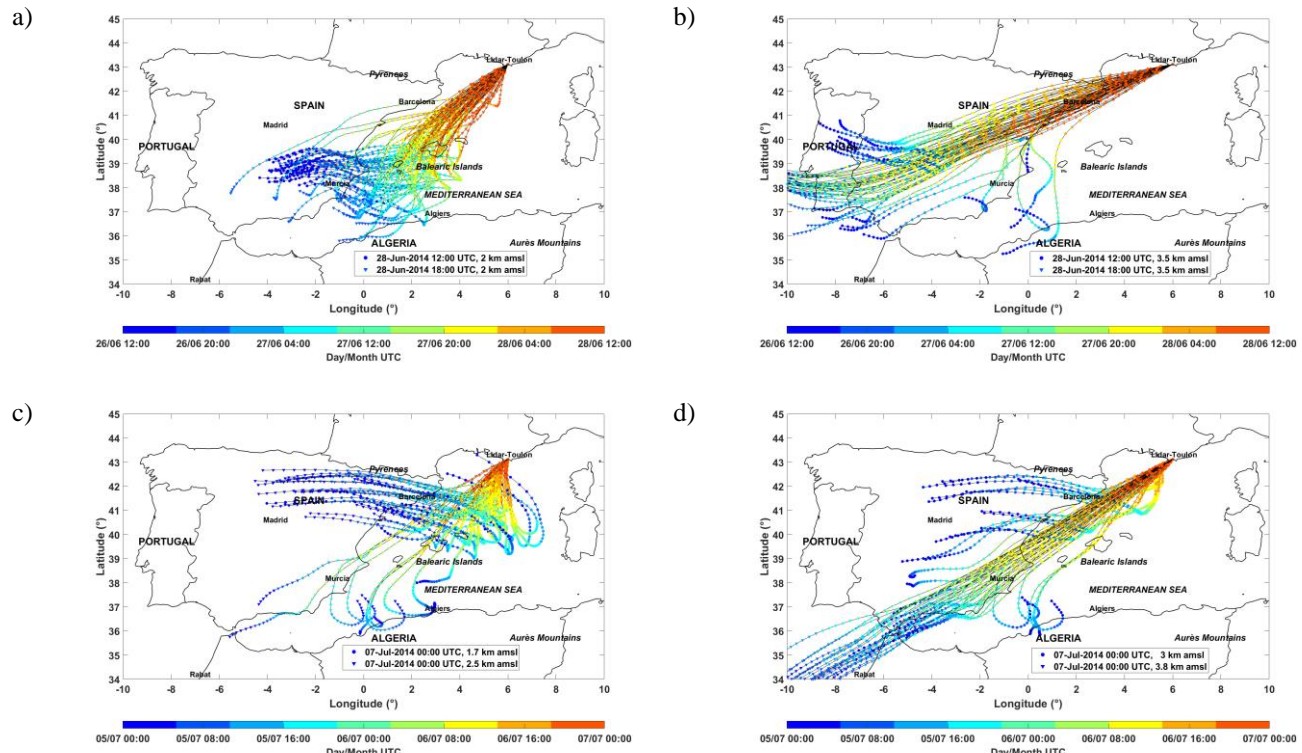

**Figure 10: Two-days back trajectories from the French Riviera (Toulon, 43.13556 ° N, 6.00944 ° E). The altitudes of the end locations of the air mass trajectories are in the main pollution plumes detected by the ground-based N₂-Raman lidar on a) 28 June, 2014 for an end location of 2 km, b) 28 June, 2014 for an end location of 3.5 km, c) 7 July, 2014 for end locations of 1.7 and 2.5 km, and d) 7 July, 2014 for end locations of 3 and 3.8 km. The Hysplit model worked in its ensemble mode, i.e. 27 back trajectories computed for each end location.**



a)

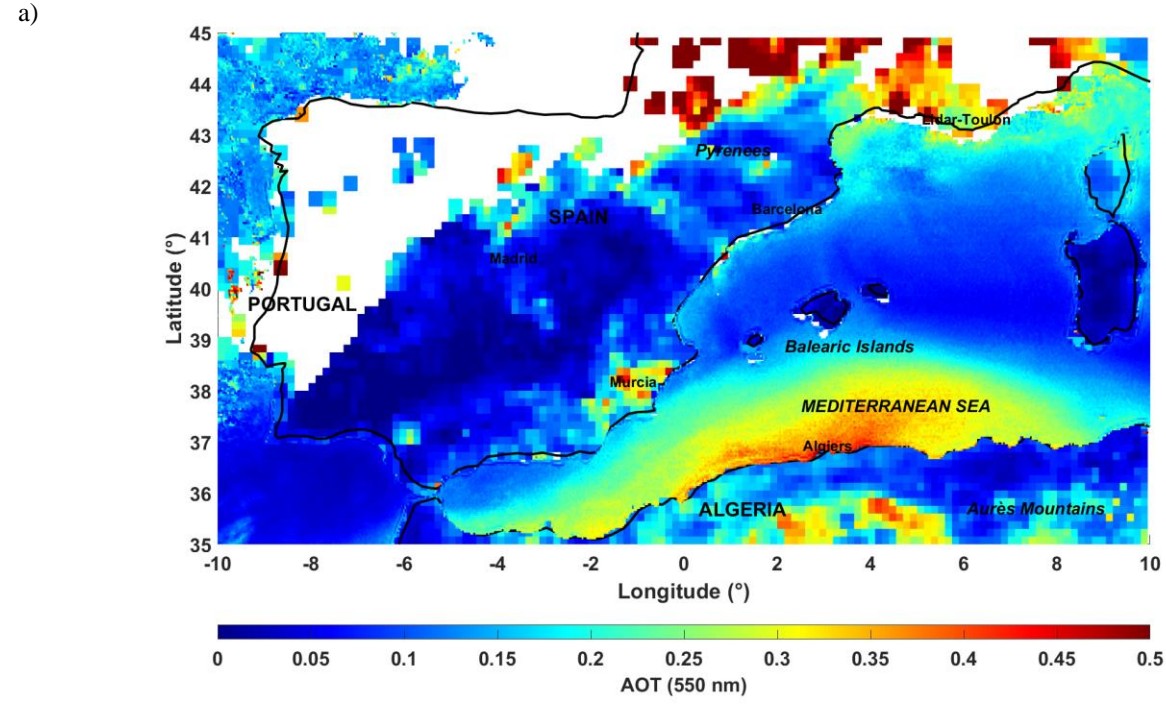

b)

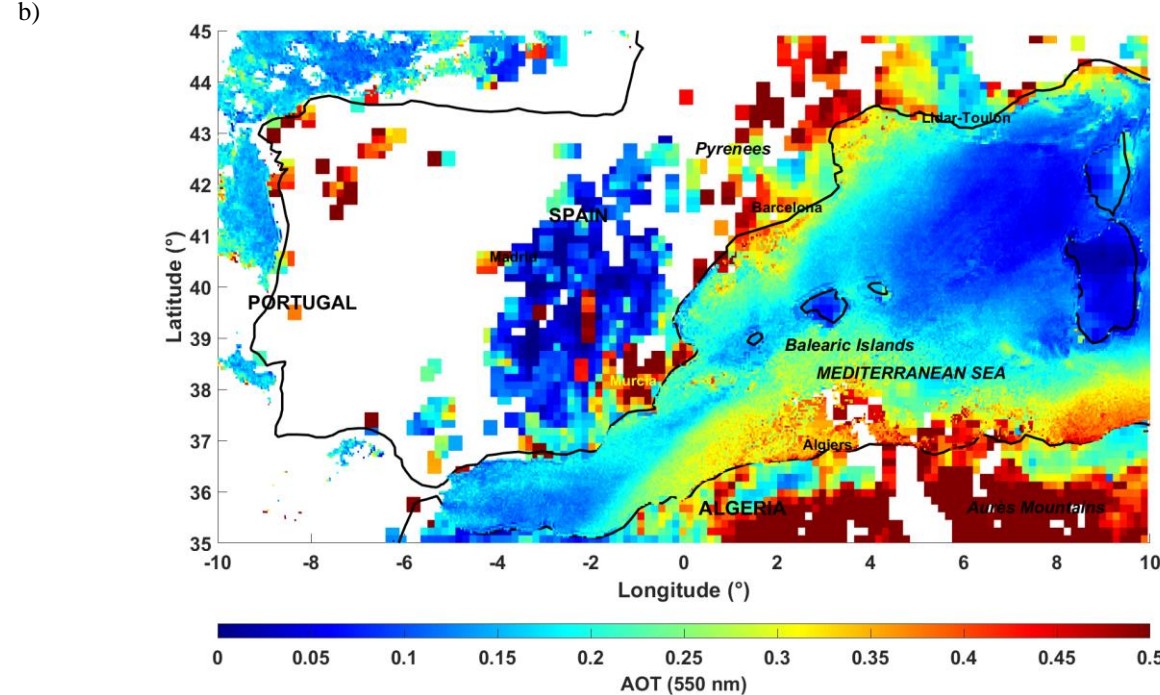

**Figure 11: AOT at 550 nm derived from the composition between MODIS over land and SEVIRI over sea observations: a) on 28 June, 2014 and b) on 6 July, 2014.**





a)

b)

**Figure 12: a) AOT at 550 nm derived from the composition between MODIS over land and SEVIRI over sea observations on 25 August, 2014. The nearest night time (~0215 UTC) ground tracks of CALIOP are given in dark-grey continuous lines. b) CALIOP-derived aerosol typing (version 4.10) as observed on August, 2014. DEM is the digital elevation model.**



## 5.2    Northern Africa contribution

Desert dust aerosol transports are rather frequent during May and June above France, and decrease in frequency over July, while remaining probable. In itself, it is not extraordinary to observe such events during a field campaign with a duration of about one month. The lifting zones located in the Sahara desert are variable, depending on the west-east travel of lows over
the Mediterranean basin (Hamonou et al., 1997). Here, we describe the two events sampled by the ground-based lidar, as highlighted by the high PDR values on Figure 2c. This will allow, among others, to evaluate the degree of coherence between the observations of the spaceborne lidar CALIOP and the ground-based lidar in the French Riviera.

The 3-day back trajectories of Figure 13 highlight the probable sources of the terrigenous aerosols, which are the Grand Erg Occidental (between Morocco and Algeria) and the Grand Erg Oriental (south-west of Tunisia). The contributions of these
two sources can be superimposed in altitude. The plumes partly sampled by the ground-based lidar are also shown on MODIS and SEVIRI AOT observations in Figure 14. They are intense events, with AOTs at 550 nm exceeding 1 along the African coast. Dispersion will decrease the AOT during transport, leading to values around 0.5 to 0.6 when reaching the French Riviera. These values match the sunphotometer measurements of the AERONET station of Toulon-La Garde on 24 June. The comparison is difficult for the second plume happening during the night of 3-4 July, however there could be an overestimation
of AOT of about 0.15 by the satellite, compared to the sunphotometer. During night time, the calculation of AOT is possible with the lidar thanks to its $N_2$-Raman channel, and yields AOT values of ~0.5 at 550 nm, when assuming a 0.8 Ångström exponent for the spectral variation of the AEC.

We have at our disposal, for each of these two dust aerosols events, a single night time CALIPSO orbit passing close to the lidar site. These orbits are traced in Figure 14. The aerosol classification given by version 4.1 of the CALIOP operational
algorithm is shown on Figure 15. Note that the plume must have moved between the observations of CALIOP and MODIS since they are 12 to 14 hours apart. However, desert dust aerosols are indeed identified, and the summit altitude of the layers is compatible with the ground-based lidar observations. The CALIOP profiles are inverted with BER values around 0.023 sr$^{-1}$ (LR = 44 sr) at 532 nm. We have found BERs (LRs) between 0.025 and 0.031 sr$^{-1}$ (32 and 40 sr) at 355 nm from the ground-based lidar profiles, which remains within the error bars.

Incidentally, Haarig et al. (2017) have not found significant differences between LR values at 355 and 532 nm for desert dust aerosols transported over Barbados, but caution is due since neither the activated sources or the transport duration are similar. Over the Balearic Islands, Chazette et al. (2016) report values of the BER (LR) between 0.020 and 0.025 sr$^{-1}$ (40 and 50 sr) for the same type of activated sources. In the synthesis table (Table 1) presented in Dieudonné et al. (2015), we note that BER (LR) values range from 0.017 to 0.029 sr$^{-1}$ (34 to 58 sr) for pure dust. Thus, there is a wide range of plausible values, which
warrants a measurement of BERs (or LRs) as often as possible so as to properly invert lidar profiles.





a)

b)

**Figure 13: Three-day back trajectories from the French Riviera (Toulon, 43.13556° N, 6.00944° E). The altitudes of the end locations of the air mass trajectories are in the main dust plumes detected by the ground-based N₂-Raman lidar on a) 24 June 2014 for end locations of 2.6, 3 and 4.5 km, b) 3 July 2014 for end locations of 3, 5 and 6 km. The Hysplit model worked in its ensemble mode, i.e. 27 back trajectories computed for each end location.**



a)

b)

**Figure 14: AOT at 550 nm derived from the composition between MODIS over land and SEVIRI over sea observations: a) is on 24 June, 2014 and b) on 3 July, 2014. The nearest night time (~0200 UTC) ground tracks of CALIOP are given in dark-grey continuous lines.**





**Figure 15:** CALIOP-derived aerosol typing (version 4.10) for the night time orbit of a) 24 June 2014 and b) 3 July 2014. The latitudinal location of the nearest latitude of the ground-based lidar is indicated by the vertical black line.



## 5.3 Other contributions

The French Mediterranean coast is a densely populated area generating trafic and industrial emissions, but also with a frequent occurrence of forest fires (Guieu et al., 2005). A rather local contribution, above 1.5 km AMSL, from the northwest is observed on July 12-13 and 15-17 in Figure 3b, which could be the result of wildfires. Indeed, their occurrence is high during this summer period in the hinterland, whose dry soils are covered mainly by garrigue, populations of cork and holm oaks, interspersed with some coniferous and palm trees. The equivalent BER of this plume is of the order of 0.022 sr$^{-1}$ and could correspond to a mixture of aerosols of biomass burning and terrigenous as has been observed over the Mediterranean, see Figure 2b by Chazette et al. (2016). Nevertheless, as for the dust aerosols, the likely BER values for biomass burning aerosols are spread over a very wide range and depend on the type of fuel, the nature of soil and the intensity of the fires, all modulated by weather situations. The forest and bush fires are fortunately mastered quickly in this region, which makes it more difficult to detect them via MODIS.

## 6    Conclusion

For the first time, a backscatter $N_2$-Raman lidar was implemented on the the French Riviera. Coupled with passive (MODIS and SEVIRI) and active (CALIPSO) spaceborne observations as well as back trajectories modelling, this instrument made it possible to identify pollution aerosol transports in the low/medium free troposphere from the Mediterranean Spanish coast to the French Riviera. Two desert aerosol transport events have also been sampled by the lidar. The likely sources of aerosol plumes trapped in the free troposphere have thus been located. So far, we had not clearly identified the contribution to the AOT of pollution plumes from the eastern Spanish coast. These can represent nearly two-thirds of the total AOT and, given their anthropogenic origins, may have a significant effect on the vertical stability of the atmosphere in the coastal area. The air masses that contain them may also subside and recirculate along the coast, which is lined with mountains.

Desert aerosols sampled by the $N_2$-Raman lidar come from the two major sources known in North-West Africa that are Grand Erg Occidental and Grand Erg Oriental. There is no noticeable difference in optical properties retrieved from the lidar measurement between these two aerosol contributions. The backscatter to extinction ratios (lidar ratios) are similar, with values of ~0.027 sr$^{-1}$ (~37 sr) at 355 nm. They are close to that derived from the CALIOP observations ($BER = 0.023$ sr$^{-1}$ or $LR = 44$ sr at 532 nm). Pollution aerosols encountered in the free troposphere above the measurement site came from the Barcelona region (Costa Brava and Costa Daurada) as well as from the Murcia region (Costa Blanca). This second origin of pollution aerosol plumes was unexpected to this extent. These two sources can be mixed before reaching the western French Riviera, but they are also separately identifiable according to the altitude. Transport altitudes are close to those of desert aerosol plumes, suggesting a similar injection process related to the large temperature difference between the sea surface and the continental air masses. Except for desert aerosols, an important point is the similarity of the BERs in the PBL and in the free troposphere. It is certainly related to the proximity of the types of aerosol production, mainly due to the traffic, but also to a low sensitivity





of the BER according to the relative humidity, even though the aerosols are hydrophilic, and a likely air mass recirculation in presence of mountain ranges close to the French Riviera coast.

This study relies on a time-limited data set (~1 month, 500 h of lidar measurements), but it raises questions as to the origin of the pollution aerosols that are sampled on the coast by the air quality stations, since the pollution may not be only local and also seems to be imported over the sea at the scale of the larger Western Mediterranean basin. It would be interesting to implement lidar systems more densely and for a longer period in the French Riviera, and further, all along Mediterranean coastlines, which are under a very strong anthropic pressure. The interest of such an approach has been shown elsewhere, where it has been found that few lidar stations were required for an air quality forecasting similar to those constrained by only ground-based in situ measurements (Wang et al., 2013). A first test was also conducted within the framework of ChArMEx using the Earlinet network (Wang et al., 2014), which we may have to complete in the upcoming years to also meet the needs of operational meteorology for forecasting extreme events.

**Acknowledgements.** The campaign was supported by the CNRS/INSU through the MISTRALS/ChArMEx & HyMeX programmes. We especially thanks F. Dulac for his help in implementing the instrumental site, J. Piazzola for his welcome and the availability of the measurement site of the Mediterranean Institute of Oceanography. This work was supported by the Commissariat à l'Energie Atomique et aux énergies alternatives (CEA). The Centre National d'Etude Spatial (CNES) helped maintain the Raman-lidar instrument. The authors would like to thank the AERONET network for sunphotometer products (at https://aeronet.gsfc.nasa.gov/). The authors acknowledge the MODIS Science, Processing and Data Support Teams for producing and providing MODIS data (at https://modis.gsfc.nasa.gov/data/dataprod/), and the Atmospheric Science Data Center (ASDC) at NASA Langley Research Center (LaRC) for the data processing and distribution of CALIPSO products (level 4.10, at https://eosweb.larc.nasa.gov/HORDERBIN/ HTML_Start.cgi). The author would like to thank the entire MSG/SEVIRI team, from ESA, Alcatel Space Industries and Matra Marconi Space. SEVIRI data have been downloaded from the ICARE Data and Services Centre (http://www.icare.univ-lille1.fr/). The NOAA Air Resources Laboratory (ARL) is acknowledged for the provision of the HYSPLIT transport and dispersion model and READY website (http://www.ready.noaa.gov) used in this publication. ECMWF data used in this study have been obtained from the ESPRI/IPSL data server.

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
