# Peer review of "Transport of aerosols over the French Riviera - Link between groundbased lidar and spaceborne observations"

_Atmospheric Chemistry and Physics, 2018_

## Referee Comment (RC1) · Anonymous Referee #2 · 30 Dec 2018

This work focuses on analyzing the vertical aerosol structure in the troposphere on the westen part of the French Riviera. In particular, the authors used a backscatter N2-Raman lidar to investigate the different aerosol properties in the boundary layer and free troposphere according the different aerosol transport patterns and source origins. In addition, satellite measurements from SEVIRI, MODIS and CALIPSO are also used to study the spacial extent of these vertical structures and properties. In cases of dust transport in the free troposphere, the authors obtain marked differences in the aerosol properties between the free troposphere and boundary layer. On the contrary, these vertical diferences are notably reduced in cases of transport of pollution particles in the free troposphere. The aim of this work, the measurements and methodology used is

an interesting approach that reveals the complexity of aerosol vertical structures in the Western Mediterranean. This should be better studied and paramatrized to improve the operational weather and air quality forecasting. In addition, the paper is well written and structured. Therefore I think it is suitable for publication in ACP.

Specific comments: P5-L9: is this native vertical sampling correct? P5-L15-L22: The two layer inversion method should be explained with more detail, since is one the novelties of the paper. P13-L22: I do not understand so well the word "supply" within the sentence "...the Sahara can supply the western Mediterranean coast.". Maybe you can change it by "reach". P17-L11: When the authors assert "The endpoints of the back trajectories are defined using the lidar profiles in Figures 2 or 3", are the endpoints of the backtrajectories referred to different altitudes or different spatial locations? please, add a comment on that. P18-L9: Can the authors add a refference justifing the sentence "..excluding desert dust 10 episodes, MODIS observations often show AOT values above 0.6 in the south of Murcia during summertime". P22-L23: Have the authors inverted the CALIOP profiles from the level 1 data? If so, pleasy especify it. If not, what inversion do the authors refer to in the sentence "The CALIOP profiles are inverted with BER values..."?

---

## Referee Comment (RC2) · Anonymous Referee #1 · 31 Dec 2018

General Comments This paper describes an observation of aerosols in French Riviera using a single-wavelength Raman lidar at 355 nm which also has a depolarization ratio measurement capability. Analysis of optical characteristics and vertical profiles of aerosols, which is possible with a single-wavelength Raman lidar, is done. Transport of aerosols is discussed with satellite data and trajectory analysis. The manuscript is generally well written. The back scatter to extinction ratio (BER) instead of the lidar ratio is used in this paper. However, nothing is simplified by using BER, even if "it is equal to the product of the single scattering albedo and of the probability of a photon being backscattered ..." It is fine to use BER, as far as both BER and lidar ratio values are indicated. However, in my opinion, it is not recommended, generally, to use BER

instead of the lidar ratio. The historical background should be respected.

Specific Comments Abstract: Wavelength of the lidar must be described before the descriptions on the BER values. Figs. 2 and 3: What are the areas indicated as "local"? Is the contribution of advection in the boundary layer not significant? From our experience, aerosols in the boundary layer can be transported quite long distance in the lower atmosphere. Though it would not be a scope of this paper, an analysis using chemical transport model would be useful to understand the emission sources and transport. The reason for insensitivity of BER to relative humidity should be discussed further, if other parameters related to particle size and refractive index are available from AERONET data. Is there any change in the depolarization ratio with relative humidity?
* * *

---

## Author Comment (AC1) · 12 Feb 2019

The comment was uploaded in the form of a supplement: https://www.atmos-chem-phys-discuss.net/acp-2018-971/acp-2018-971-AC1-supplement.pdf

---

## Author Comment (AC2) · 12 Feb 2019

**Response to reviewers**

Dear Editor, please find hereafter the response to the referee's comments. We thank the reviewers for thoughtful and constructive proposals on our manuscript. We appreciate the time they invested in the review. We believe that our revised manuscript addresses all the comments.

In the following, the comments made by the referees appear in black italic, our replies are in bold, and the proposed modified text in the manuscript is in blue.

**Reviewer #1**

*P5-L9: is this native vertical sampling correct?*

**Yes, the native vertical sampling of the lidar is 0.75 m. We have modified the sentence by "The signal acquisition is performed using the PXI technology, manufactured by the National Instruments$^{TM}$ company (http://www.ni.com), and using an acquisition board with a sampling rate of 200 MHz corresponding to a lidar native vertical sampling of 0.75 m."**

*P5-L15-L22: The two layer inversion method should be explained with more detail, since is one the novelties of the paper.*

**The method used is very similar to that with a single layer, well described within the references given in the paper. This is not a very original approach. We reviewed the text to be clearer:**

**"This method builds on the case of a single aerosol layer described above. We begin by inverting the upper layer and determine its BER or LR, then we apply the same approach to the lower layer, retaining the LR of the higher layer. The constraint is given by the partial AOTs (Dieudonné et al., 2015) calculated from the N2-Raman channel for each of the aerosol layers previously located in altitude. The transition altitude between the layers is determined manually and the continuity is ensured by a sigmoid function on a thickness of about 1 km between the two layers. Such an approach is possible, just like in the case of a single aerosol layer, if the SNR is larger than 10. This leads us to produce night time profiles with a time average of 5 hours, between 23:00 and 04:00."**

*P13-L22: I do not understand so well the word "supply" within the sentence "...the Sahara can supply the western Mediterranean coast.". Maybe you can change it by "reach".*

**Yes, the correction has been done.**

*P17-L11: When the authors assert "The endpoints of the back trajectories are defined using the lidar profiles in Figures 2 or 3", are the endpoints of the backtrajectories referred to different altitudes or different spatial locations? please, add a comment on that.*

**We have specified by adding: "to determine both their temporal and altitude locations above the lidar".**

*P18-L9: Can the authors add a refference justifing the sentence "..excluding desert dust 10 episodes, MODIS observations often show AOT values above 0.6 in the south of Murcia during summertime".*

**We have not specific references for that. It is the observations of MODIS data on the period of interest. The text was rewrite for the sake of clarification: "This area can therefore be a strong emitter of aerosol precursors. Without consider the desert dust episodes, we checked that MODIS observations often show AOT values above 0.6 in the south of Murcia during summertime (https://worldview.earthdata.nasa.gov)."**

*P22-L23: Have the authors inverted the CALIOP profiles from the level 1 data? If so, pleasy especify it. If not, what inversion do the authors refer to in the sentence "The CALIOP profiles are inverted with BER values..."?*

**We used the operational product that are well validated. This point has been clarified by adding "The CALIOP products version 4.10 give BER…".**

**Reviewer #2**

*The back scatter to extinction ratio (BER) instead of the lidar ratio is used in this paper. However, nothing is simplified by using BER, even if "it is equal to the product of the single scattering albedo and of the probability of a photon being backscattered ..." It is fine to use BER, as far as both BER and lidar ratio values are indicated. However, in my opinion, it is not recommended, generally, to use BER instead of the lidar ratio. The historical background should be respected.*

**It's the same thing to use the BER or the LR. The BER is simply a little more directly related to the radiative transfer equations. In this article, we often give the two so that everyone can navigate. We used the BER for historical reasons and our algorithms are based on it. In our opinion, this does not detract from scientific quality or clarity.**

*Abstract: Wavelength of the lidar must be described before the descriptions on the BER values.*

**Yes, it is true. We have made the correction: "For the first time, a 355 nm backscatter $N_2$-Raman lidar has been deployed…".**

*Figs. 2 and 3: What are the areas indicated as "local"? Is the contribution of advection in the boundary layer not significant? From our experience, aerosols in the boundary layer can be transported quite long distance in the lower atmosphere. Though it would not be a scope of this paper, an analysis using chemical transport model would be useful to understand the emission sources and transport.*

**We have never observed long-distance transport in the boundary layer. They do not exceed a few hundred kilometers. In the particular case of this article, as explained, we are in the coastal regime with alternating sea/continental breezes associated with an ascendance of the air masses when they pass over the continent. At the origin of the air**

masses, it is the opposite, they pass from a hot surface to a cold sea. They are therefore be located above the marine boundary layer during transport. All this does not plead for a long-distance transport in the marine boundary layer. The "local" term is therefore related to nearby local or regional aerosol emissions as explained for forest fires. We have modified the sentence of section 5.3: "The French Mediterranean coast is a densely populated area generating trafic and industrial emissions, identified by the "local" indicator in Figures 2b and 3b, but also with a frequent occurrence of forest fires (Guieu et al., 2005)."

*The reason for insensitivity of BER to relative humidity should be discussed further, if other parameters related to particle size and refractive index are available from AERONET data. Is there any change in the depolarization ratio with relative humidity?*

Such an aspect has been studied in the previous paper of Raut and Chazette (2008b) cited in the text, and we use their results in section 3.2. They highlighted that the relative humidity does not significantly affect the BER of traffic aerosols. As we write in section 3.2, the low vertical variability of the BER observed during the field experiment support the previous findings. In our case, it is difficult to go further because profiles of the chemical composition and the size distribution of aerosol are necessary. These types of measurements are only available via airborne measurements, which are difficult to perform above inhabited area.